# Organic Matter Transformations are Disconnected Between Surface Water and the Hyporheic Zone

James C. Stegen[1], Sarah J. Fansler [1], Malak M. Tfaily[4], Vanessa A. Garayburu-Caruso[1], Amy E. Goldman[3], Robert E. Danczak[1], Rosalie K. Chu[2], Lupita Renteria[1], Jerry Tagestad[3], Jason Toyoda[2]

[1]Earth and Biological Sciences Directorate, Pacific Northwest National Laboratory, Richland, WA, U.S.A; james.stegen@pnnl.gov (J.C.S.); sarah.fansler@pnnl.gov (S.J.F.); vanessa.garayburu-caruso@pnnl.gov (V.A.G-C.); robert.danczak@pnnl.gov (R.E.D.); lupita.renteria@pnnl.gov (L.R.);

[2]Environmental Molecular Sciences Laboratory, Richland, WA 99352, USA; Rosalie.Chu@pnnl.gov (R.K.C.);

Jason.Toyoda@pnnl.gov (J.T.)

[3]Energy and Environment Directorate, Pacific Northwest National Laboratory, Richland, WA, U.S.A; amy.goldman@pnnl.gov (A.E.G.); Jerry.Tagestad@pnnl.gov (J.T.)

[4]Department of Environmental Science, University of Arizona, Tucson, AZ, 85719, USA; tfaily@email.arizona.edu (M.M.T.)

*Correspondence to*: James C. Stegen (James.Stegen@pnnl.gov)

**Abstract**

Biochemical transformations of organic matter (OM) are a primary driver of river corridor biogeochemistry, thereby modulating ecosystem processes at local to global scales. OM transformations are driven by diverse biotic and abiotic processes, but we lack knowledge of how the diversity of those processes varies across river corridors and across surface and subsurface components of river corridors. To fill this gap we quantified the number of putative biotic and abiotic transformations of organic molecules across diverse river corridors using ultra-high resolution mass spectrometry. The number of unique transformations is used here as a proxy for the diversity of biochemical processes underlying observed profiles of organic molecules. For this, we use public data spanning the contiguous United States (ConUS) from the Worldwide Hydrobiogeochemical Observation Network for Dynamic River Systems (WHONDRS) consortium. Our results show that surface water OM had more biotic and abiotic transformations than OM from shallow hyporheic zone sediments (1-3cm depth). We observed substantially more biotic than abiotic transformations, and the number of biotic and abiotic transformations were highly correlated with each other. We found no relationship between the number of transformations in surface water and sediments, and no meaningful relationships with latitude, longitude, or climate. We also found that the composition of transformations in sediments was not linked with transformation composition in adjacent surface waters. We infer that OM transformations represented in surface water are an integrated signal of diverse processes occurring throughout the upstream catchment. In contrast, OM transformations in sediments likely reflect a narrower range of processes within the sampled volume. This indicates decoupling between the processes influencing surface water and sediment OM, despite the potential for hydrologic exchange to homogenize OM. We infer that the processes influencing OM transformations and the scales at which they operate diverge between surface water and sediments.

## 1 Introduction

River corridors are an important component of the integrated Earth system that have large influences on the flux of materials and energy across local to global scales (Harvey and Gooseff, 2015; Schlünz and Schneider, 2000; Schlesinger and Melack, 1981). The biogeochemical function of river corridors (e.g., rates of contaminate transformations) are the outcome of both biotic and abiotic processes (e.g., He et al., 2016; Bowen et al., 2020). On the biological side, microbial communities in areas where groundwater and surface water mix (i.e., hyporheic zones) can, for example, contribute substantially to river corridor respiration rates (Jones Jr, 1995; Naegeli and Uehlinger, 1997; Battin et al., 2003; Fischer et al., 2005; but see Ward et al., 2018). In these areas, microbial metabolism can be heavily modified by hydrologic mixing (e.g., McClain et al., 2003; Stegen et al., 2016, 2018). On the abiotic side, light-driven organic matter (OM) transformations, for example, can consume significant amounts of dissolved organic carbon in river systems (e.g., Amon and Benner, 1996) and heavily modify OM profiles (e.g., Holt et al., 2021). The integration of biotic and abiotic processes ultimately lead to variation in water quality and ecosystem fluxes that are relevant to local communities and global fluxes.

Within river corridors, OM serves as a primary energy source fueling aerobic and anaerobic heterotrophic respiration (Fisher and Likens, 1973; Wetzel, 1995; Cole et al., 2007; Creed et al., 2015). The chemistry of OM in river corridors is particularly important, with a multitude of influences over biogeochemical rates and ecosystem fluxes. For example, through field, lab, and mechanistic modeling, thermodynamic properties of OM have been shown to influence microbial respiration in both aerobic and anaerobic river corridor settings (Boye et al., 2017; Stegen et al., 2018; Graham et al., 2018; Garayburu-Caruso et al., 2020a; Song et al., 2020; Sengupta et al., 2021). This has also recently been shown in soil systems as well (Hough et al., 2021). Other attributes of OM chemistry, such as the carbon to nitrogen ratio, also have strong influences over river corridor rates/fluxes (Bauer et al., 2013; Liu et al., 2020). As is the case for nearly all attributes of river corridors, the spatial variation in and temporal dynamics of OM chemistry emerge through the integration of biotic and abiotic processes.

Biotic and abiotic processes influence river corridor OM chemistry by modifying rates of production, transformation, sorption/desorption, and/or spatial movement (Danczak et al., 2020). All these factors have been studied to some degree in river corridors, and advances in cheminformatics techniques can provide further insights specifically into the biotic and abiotic components of OM transformations. More specifically, Fudyma et al. (2021) used the ultra-high mass resolution of Fourier transform ion cyclotron resonance mass spectrometry (FTICR-MS) data (Marshall et al., 1998; Bahureksa et al., 2021) to infer putative abiotic and abiotic transformations of OM in a river corridor system. This extended previously-developed cheminformatics techniques (e.g., Breitling et al., 2006; Stegen et al., 2018; Danczak et al., 2020, 2021) to include abiotic transformations. Fudyma et al. (2021) found that abiotic OM transformations, such as those driven by sunlight and photooxidation, may alter bioavailability of OM in groundwater and surface water. These observations were collected across different subsurface hydrologic mixing conditions and suggest that changes in the bioavailability of OM lead to enhanced microbial activity in subsurface domains like the hyporheic zone. This emphasizes the need to consider abiotic OM transformations as a key

complement to biotic OM transformations in river corridors (Amon and Benner, 1996; Bowen et al., 2020; Holt et
al., 2021; Hu et al., 2021).

While both biotic and abiotic OM transformations are important in river corridors, we lack broad cross-system
understanding of how these two classes of transformations relate to each other and how they vary between hyporheic
zone sediments and surface water. Resolving these knowledge gaps is useful from a number of perspectives; for
example, it was recently proposed that surface water chemistry can be used as a mirror to understand subsurface
chemistry and associated processes (Stewart et al., 2021). With that idea in mind, if transformation numbers or
profiles in surface water are statistically associated with transformation numbers or profiles in sediments, we could
use surface water data (easier to generate) to infer properties/processes in the subsurface (much harder to study). In
addition, such correspondence would indicate that surface-subsurface hydrologic exchange in river corridors is
sufficient to overcome localized processes, thereby at least partially homogenizing OM across river corridor
compartments. On the other hand, lack of correspondence between surface water and sediment OM transformations
would indicate that deterministic processes (*sensu* Danczak et al., 2020) in the subsurface overwhelm transport
mechanisms in governing OM chemistry. Either outcome is highly informative for fundamental understanding and
for mechanistic modeling efforts that couple surface-subsurface hydrology and biogeochemistry (e.g.,
hyporheicFoam Li et al., 2020).

Here we aim to help fill knowledge gaps associated with OM transformation counts and composition across surface
and subsurface components of river corridors distributed across the contiguous United States (ConUS). We
specifically compare the numbers of biotic and abiotic OM transformations in sediments and surface waters, and
evaluate the potential for continental-scale spatial patterns in biochemical transformation counts and composition.
To do so, we use publicly available FTICR-MS data provided by the Worldwide Hydrobiogeochemistry Observation
Network for Dynamic River Systems (WHONDRS) consortium (Stegen and Goldman, 2018). One key outcome of
our analyses is that OM transformations in sediments are not related to OM transformations in adjacent surface
water, which suggests divergent governing processes despite hydrologic connectivity between these river corridor
sub-systems.

**2 Methods**
*Data Generation*
The samples used for data generation were collected and processed in 2019 as part of the WHONDRS consortium
(Stegen and Goldman, 2018), and the data were retrieved from publicly available data packages (Toyoda et al.,
2020; Goldman et al., 2020). Full details on sample and metadata collection are provided in Garayburu-Caruso et al.
(2020b); some additional sample data are used here that were not used in Garayburu-Caruso et al. (2020b), but all
methods are consistent. In short, at each site (Fig. 1) three depositional zones within ~10 m of each other were
sampled for shallow sediments (~1-3cm into the riverbed). Prior to sediment collection, surface water was collected
at the most downstream sediment sampling location. The samples were shipped to the Pacific Northwest National
Laboratory (PNNL) campus in Richland, WA (USA) on blue ice within 24 hours of collection. Untargeted
characterization of OM was done using ultrahigh resolution FTICR-MS. In preparation for FTICR-MS analysis,
sediments were extracted with Milli-Q deionized (DI) water and the resulting supernatant was filtered prior to
measurement of non-purgeable organic carbon (NPOC). NPOC concentrations were normalized to 1.5 mg C L$^{-1}$ by
adding Milli-Q DI water. To remove salts and minerals, 15 ml of each sample were then passed through PPL
cartridges (Bond Elut). FTICR-MS analyses were performed at the Environmental Molecular Science Laboratory
(EMSL) in Richland, WA using a 12 Tesla Bruker SolariX FTICR mass spectrometer (Bruker, SolariX, Billerica,
MA, USA) in negative ionization mode. FTICR-MS spectra were processed to assign molecular formulae as
described in Garayburu-Caruso et al. (2020b). Briefly, to convert raw FTICR-MS spectra into a list of mass-to-
charge ratios (i.e., m/z values) we used BrukerDaltonik Data Analysis (version 4.2). We specifically applied the
FTMS peak picker module with a signal-to-noise ratio (S/N) threshold of 7 and absolute intensity threshold of 100.
We then used Formularity (Tolić et al., 2017) to align peaks with a 0.5 ppm threshold and assign chemical formulas.
Within Formularity we specifically used the Compound Identification Algorithm with S/N > 7 and mass
measurement error of <0.5 ppm. The Compound Identification Algorithm algorithm allows for C, H, O, N, S, and P
within the assigned formula, while excluding other elements.

FTICR-MS data were used as presence-absence due to peak intensities providing unreliable estimates of absolute or
relative concentrations, which is a limitation inherent to FTICR-MS analysis. While FTICR-MS provides the most
comprehensive OM chemistry characterization currently available, it has constraints such as not being quantitative
and missing low molecular weight compounds ( ~ <200 Da) that need to be taken into consideration. FTICR-MS
nonetheless provides a robust approach for conducting untargeted characterization of environmental OM.

In addition to the FTICR-MS data, we used a suite of environmental variables in an attempt to explain variation in
OM transformation counts. These variables included actual evapotranspiration, mean annual precipitation, mean
annual temperature, and potential evapotranspiration. Global datasets for these variables were acquired from two
sources as geospatial raster datasets: The historical mean annual temperature and mean annual precipitation were
downloaded from worldclim.org (Fick and Hijmans, 2017) and the evapotranspiration and potential
evapotranspiration were available as geospatial rasters from the MOD16 Global Evapotranspiration Product
database (Running et al., 2017). The environmental variable values were associated with each sample location using
ArcGIS function *Extract Values to Points*. The output was a table of climate and evapotranspiration values for each
sample location.

*Biochemical transformation analyses and statistics*
Biochemical transformations of OM were inferred as in Fudyma et al. (2021), and full details of the method can be
found in that publication. In brief, we used a list of common biochemical transformations (see file 'Biotic-abiotic-
transfromation-classification.csv' in the Stegen et al. (2021) data package) to putatively infer the identity (e.g.,
hydrogenation, loss/gain of an alanine, etc.) and number of occurrences of each transformation in each sample. A
given transformation was inferred each time we observed the corresponding mass shift between a pair of peaks,
within each sample. This analysis does not provide direct information about where or when a given transformation
may have occurred, and it is likely that they occurred prior to the sample being taken and outside of the sampled
volume. For example, surface water acts as an integrator whereby transformations inferred in surface water samples
likely occurred throughout the upstream catchment. What is observed in surface water samples is therefore the
cumulative result of processes throughout the upstream catchment. Similarly, biochemical transformations inferred
from sediment samples may have occurred along subsurface flow paths beyond the sampled volume.

In each sample, we counted the number of times each transformation was inferred to have occurred. We then
designated each transformation as biotic, abiotic, or both reflecting the potential chemical reaction sources as in
Fudyma et al. (2021). Next, the samples were parsed into sediment or surface water categories. Then we compared
the total number of transformations, the number of abiotic transformations, the number of biotic transformations,
and the ratio of abiotic to biotic transformation numbers for each sample. Distributions based on the number of
transformations or their ratio were compared between surface water and sediments using Wilcox signed rank tests.
Transformation numbers and their ratio were related to each other and to spatial and environmental variables using
ordinary least squares regression. Spatial and environmental variables included latitude, longitude, and the
environmental variables listed above.

In addition to studying transformation numbers, we examined the composition of transformations and related these
compositional profiles between surface water and sediments. The purpose of this analysis was to evaluate the degree
to which hydrologic exchange homogenizes OM between sediments and physically adjacent surface water. The
compositional profile for each sample was characterized by the number of times each transformation was inferred.
For each site, the three surface water samples were combined by adding together the number of observations for
each transformation and then computing the relative abundance of each transformation. The same process was done
for the three sediment samples within each site. Doing this across all sites provided the equivalent of an ecological
'species-by-site' matrix, but with transformations as 'species' and samples as 'sites' and the entries as the site-level
relative abundance of each transformation in each sample. In turn, we calculated Bray-Curtis dissimilarity among all
sediment samples and, separately, among all surface water samples. The relationship between surface water and
sediment Bray-Curtis dissimilarities was then evaluated using distance-matrix regression and a Mantel test to
account for non-independence of the pairwise comparisons. For this, the Bray-Curtis values from surface water from
a given site were linked with the Bray-Curtis values for the sediment data from the same site. Each data point used
in the regression is therefore based on surface water and sediment from the same site compared to data from a
different, but common, site. For example, in the case of three sites (A, B, and C), a single data point in the regression
would be based on water from A compared to water from B and sediments from A compared to sediments from B.
Another data point would be water from A compared to water from C and sediments from A compared to sediments
from C, and so on. If hydrologic transport between surface water and sediments homogenizes organic molecules
between water and sediments, water Bray-Curtis should increase with sediment Bray-Curtis. The stronger the
homogenization, the stronger the Bray-Curtis relationship should be. If hydrologic transport does not homogenize
OM between sediments and the physically adjacent surface water, no relationship will be observed between surface
water and sediment Bray-Curtis values.

**3 Results and Discussion**
Examining ConUS-scale distributions for the number of putative biotic and abiotic transformations showed that
surface water OM had significantly more biotic ($W = 12360$, $p \ll 0.0001$, Fig. 2A) and abiotic ($W = 12978$, $p \ll$
$0.0001$, Fig. 2B) transformations than sediment OM. In addition, there were many fewer abiotic transformations
(~50-800 per sample) than biotic transformations (~5000 to 80000) within the ConUS-scale distributions (cf., Fig.
2A,B). On a per-sample basis the abiotic to biotic ratio ranged from ~0.01 to 0.02, and sediments had a significantly
higher ratio than surface water ($W = 46627$, $p \ll 0.0001$, Fig. 2C). As a key methodological detail--as described in
the Methods section--we note that all samples were normalized to a constant organic carbon concentration prior to
FTICR-MS analysis such that comparisons can be made directly among all samples, including between surface
water and sediments.

The larger number of putative biotic and abiotic transformations in surface water is, at first, surprising given that
hyporheic zone sediments are very biogeochemically active (Naegeli and Uehlinger, 1997; McClain et al., 2003),
and are often considered as ecosystem control points within river corridors (Bernhardt et al., 2017). We might
therefore expect there to be more OM transformations in hyporheic zone sediments. It is important to consider,
however, that the number of transformations (as quantified here) is a reflection of transformation diversity, not the
rate of OM transformations. For example, a system may experience a very high rate of OM transformation, but have
a low number of unique types of transformations. Such a situation would result in a low transformation count due to
the FTICR-MS data being used to indicate the presence or absence of organic molecules (i.e., there is no information
on abundance).

Given that the number of putative transformations does not indicate the rate of transformation, the larger number in
surface water may result from surface water OM being an integrated signature of processes occurring across
upstream catchments (Vannote et al., 1980; Xenopoulos et al., 2017). In comparison, sediment OM may reflect
processes occurring within and/or much closer to the sampled volume. That is, a larger diversity of transformations
may accumulate as surface water OM integrates processes and sources from across the stream network, which is
conceptually consistent with previous work using the same data that found higher molecular richness in surface
water than in sediment OM (Garayburu-Caruso et al., 2020b). This highlights that inferred transformations likely
occurred prior to sampling and outside of the sampled volume (e.g., in the upstream catchment for surface water
data and along subsurface flow paths for sediment data). Our interpretation furthermore sets up the emergent (i.e.,
*post-hoc*) hypothesis that the number of transformations may increase with catchment area. This hypothesis could be
evaluated by combining the dataset analyzed here with quantification of upstream catchment areas. Furthermore,
this points to a need to compare drivers of transformation counts with drivers of OM functional diversity. For
example, Kida et al. (2021) recently found OM functional diversity to increase, decrease, or stay steady moving
down a stream network (i.e., as upstream catchment area increased). Those authors tied variability in the patterns to
context dependencies in environmental characteristics. ConUS-scale consistency in the patterns observed here for
OM transformation contrasts with the context dependencies observed for OM functional diversity in Kida et al.
(2021). We therefore encourage future studies to elucidate relationships between OM transformations and functional
diversity.

While the number of abiotic transformations was far less than biotic transformations both locally (i.e., within each
site) and at the ConUS-scale (Fig. 2), abiotic transformations nonetheless play an important role in river corridors
(Judd et al., 2007; Ward et al., 2017). For example, Fudyma et al. (2021) examined biochemical transformations in
the river corridor and found that abiotic transformations in surface water modified the chemistry of OM entering the
hyporheic zone, with subsequent impacts to respiration rates. Soares et al. (2019) also recently found that abiotic
transformations of OM can lead to increases in bioavailable OM as residence time of surface water increases. These
demonstrations of the importance of abiotic transformations further emphasize that the number of transformations
observed here is a quantification of transformation diversity, not functional importance. That is, small sets of
transformations can serve vital functional roles and can connect sets or 'modules' of transformations together
(Fudyma et al., 2021).

As noted above, our results suggest that OM transformations in surface water may reflect processes occurring across
the upstream catchment while OM transformations in sediment may reflect processes within the sampled volume.
This inference was further supported by non-significant relationships between surface water and sediments in terms
of transformation counts (Fig. 3). That is, the number of abiotic transformations in surface water was not related to
the number of abiotic transformations in sediments. This analysis was done on paired samples, with data for surface
water coming from the same stream reach as data for sediments. This allowed for regression-based analyses. The
number of biotic transformations and the abiotic-to-biotic ratio were also uncorrelated between surface water and
sediments. Extending the analyses to transformation composition further supports a disconnect between surface
water and sediment OM transformation profiles. That is, we observed no meaningful relationship between surface
water and sediment OM transformation compositional dissimilarity (Figs. 4, S1). As discussed in the Methods
section, if hydrologic transport was overwhelming localized processes, we would have observed a clear positive
relationship. Instead, a very weak relationship was observed ($R^2 = 0.04$), indicating that influences of transport are
very small relative to localized processes. This may be conceptualized similarly to the Damköhler number whereby
the ratio of the reaction-influence to the transport-influence is very large.

The lack of correlation between transformation counts and composition between surface water and sediment OM
indicate at least a partial decoupling of the processes governing OM transformations in surface water and sediments.
In this case, bi-directional exchanges (i.e., hyporheic exchange) (Harvey and Gooseff, 2015) of water and OM
between surface water and the sediments are not strong enough to overwhelm processes occurring within each
subsystem. It was recently proposed that OM assemblages can be thought of in terms of ecological community
assembly processes including stochastic dispersal and deterministic selection (Danczak et al., 2020, 2021). From this
ecological perspective, our results indicate that the rate of dispersal (i.e., transport) of OM from surface water into
sediments is not sufficient to overcome the influences of localized, deterministic processes that cause systematic
differences (among molecules) in the rates of production and transformation. Here, OM production and
transformation are analogous to organismal birth and death, respectively (Danczak et al., 2020). It is unclear,
however, what factors and processes within the sediments impose deterministic selection over molecular production
and transformation. We hypothesize that a suite of factors are at work, such as redox conditions and sediment
mineralogy. For example, the profile of organic molecules can be influenced by sorption, desorption, and
transformations associated with organo-mineral interactions (Mead and Goñi, 2008; Zhou and Broodbank, 2014; Le
Gaudu et al., 2022). It is also plausible that lower OM diversity in sediments, relative to surface water (Garayburu-
Caruso et al., 2020b), could be due to organo-mineral interactions selecting for and against certain types of organic
molecules (Aufdenkampe et al., 2007; Kleber et al., 2007, 2021). It is these kinds of localized interactions that we
propose as overcoming strong coherence between surface water and sediment OM that may otherwise occur via
transport and mixing. and mixing over effectively Spatial variation in mineralogy, redox, and other physicochemical
properties may therefore help explain variation across sediments in the number of observed transformations.

In contrast to the decoupling between OM transformations in surface water and sediments, we observed strong
correlations between the number of biotic and abiotic transformations within surface water and within sediment
(Figure 5). As discussed above, the number of transformations is best interpreted as a measure of transformation
richness, as opposed to an indication of rates. The strong correlation between biotic and abiotic transformation
counts therefore indicates that the diversity of biotic transformations tracks closely with the diversity of abiotic
transformations. This suggests that systems in which a larger range of biochemical mechanisms contribute to OM
production and transformation are also characterized by a larger range of abiotic mechanisms contributing to OM
transformations. In considering this inference, it is important to recognize that the correlation between biotic and
abiotic transformation counts may be influenced by among-sample variation in the number of observed molecules.
However, among-sample variation in the number of observed molecules is not an artifact. This is because higher
OM transformation richness should lead to a larger number of unique organic molecules. That is, the number of
observed molecules and the level of OM transformation richness are mechanistically linked to each other whereby
richness can beget more richness. This lends credence to our inferences above, but also emphasizes that additional
insights can be gleaned by controlling for among-sample variation in the number of observed molecules.

To control for among-sample variation in the number of observed molecules we quantified the within-site abiotic-to-
biotic ratio. This ratio was significantly higher in sediments than in surface water. The close spatial proximity
between OM and mineral surfaces in sediments may contribute to relatively higher frequency of abiotic
transformations in sediments. This may be associated, in part, with sorption/desorption processes (Kleber et al.,
2021), though OM compositional change associated with desorption in the hyporheic zone can be strongly linked to
microbially-mediated transformations (Zhou et al., 2019). In addition, a larger diversity of redox conditions and thus
more diverse redox species (Briggs et al., 2013; Boano et al., 2014; Lewandowski et al., 2019) in sediments could
also contribute to the larger relative contribution of abiotic transformations in sediments. This does not discount the
important role of abiotic transformations in surface water, such as those associated with photooxidation. Indeed, it is
well known that abiotic transformations in surface water can strongly influence watershed carbon cycling fluxes
(Ward et al., 2017; Bowen et al., 2020; Hu et al., 2021).

In addition to comparing transformations across river corridor subsystems, we conducted a preliminary investigation
of spatial and climate correlates (e.g., mean annual temperature) of transformation numbers. This revealed non-
significant ($p > 0.05$) or very weak ($R^2 < 0.1$) relationships in all cases (see Supplementary Figures). We also
performed multiple regression analyses and even models with 5 spatial and climate variables showed very low
explanatory power (e.g., $R^2 < 0.08$ for the model explaining variation in total transformations). Low explanatory
power of space and climate is surprising given continental-scale variation in OM chemistry revealed in the same
dataset used here. That is, Garayburu-Caruso et al. (2020b) found a significant increase in sediment mean nominal
oxidation state of organic carbon (NOSC) in the eastern US, relative to the western US. The lack of relationships
shown here indicates that large-scale drivers of OM chemistry are not the same factors that drive variation in the
number of transformations or the abiotic-to-biotic transformation ratio. A major remaining challenge is, therefore, to
elucidate what drives variation in the absolute and relative numbers of abiotic and biotic OM transformations, and
understand relationships between transformations and functional diversity of attributes such as NOSC.

**5 Conclusions**
While it is unclear what drives variation in transformation numbers across river corridors, our ConUS-scale analyses
provided insights that are likely applicable across all river corridors. In particular, processes governing OM
transformations appear to be distinct between surface water and hyporheic zone sediments. This is unexpected given
the bidirectional exchange of materials between surface water and sediments (Boano et al., 2014; Harvey and
Gooseff, 2015). It also highlights that while hydrologically-driven mixing can stimulate biogeochemical processes in
hyporheic zones (McClain et al., 2003; Stegen et al., 2016), it generally does not homogenize OM between surface
water and sediments (Stegen et al., 2018; Fudyma et al., 2021). Instead, we propose that OM observed in each
subsystem is the result of biochemical transformations mediated by distinct processes. We emphasize that this
inference extends only to the analytical limits of the FTICR-MS data used here, which does not provide a
comprehensive survey of all possible transformations. However, no analytical method can provide a comprehensive
survey. Among currently available methods, FTICR-MS provides the highest resolving power to enable the most
comprehensive non-targeted surveys of organic molecules in environmental samples (Bahureksa et al., 2021). As
such, using additional methods (e.g., liquid chromatography-MS) will increase the number of putative
transformations inferred in each sample, but the total number of transformations should be dominated by those
inferred from FTICR-MS data. We encourage use of multiple complementary methods in future studies, as this can
be a powerful approach (Kim et al., 2006; Hagel and Facchini, 2008; Wolfender et al., 2015; Wilson and Tfaily,
2018; Kamjunke et al., 2019; Tfaily et al., 2019). We hypothesize, however, that using multiple methods will not
modify our primary inference. That is, surface OM transformation counts are likely influenced by upstream
catchment processes while sediment OM is likely influenced by processes local to the sample volume. These
observations further highlight the need to study and model river corridors through a multi-scale perspective.

**6 Code availability:** Scripts to reproduce the primary results of this manuscript are available in Stegen et al. (2021).

**7 Data availability:** Data to reproduce the primary results of this manuscript are available in Stegen et al. (2021)**.**
The data were retrieved from published data packages (Toyoda et al., 2020; Goldman et al., 2020).

**8 Author contributions:** JCS (Conceptualization, Formal Analysis, Funding acquisition, Investigation,
Methodology, Project administration, Software, Supervision, Validation, Visualization, Writing – original draft
Writing – review & editing), SJF (Conceptualization, Formal Analysis, Investigation, Methodology, Software,
Validation, Visualization, Writing – original draft, Writing – review & editing),  MMT (Conceptualization,
Investigation, Methodology, Writing – review & editing), VAG-C (Data curation, Investigation, Writing – review &
editing), AEG (Data curation, Investigation, Writing – review & editing), RED (Data curation, Investigation,
Software, Writing – review & editing), RKC (Data curation, Investigation, Writing – review & editing), LR (Data
curation, Investigation, Writing – review & editing), JeT (Data curation, Investigation, Writing – review & editing),
JaT (Data curation, Investigation, Writing – review & editing)

**9 Competing interests:** The authors declare that they have no conflict of interest.

**10 Acknowledgements**
This work was supported by the U.S. Department of Energy (DOE) Office of Science Early Career Research
Program at Pacific Northwest National Laboratory (PNNL). PNNL is operated by Battelle for the U.S. DOE under
Contract DE-AC05-76RL01830. This study used data from the Worldwide Hydrobiogeochemistry Observation
Network for Dynamic River Systems (WHONDRS) under the River Corridor Science Focus Area (SFA) at PNNL.
The SFA is supported by the U.S. DOE, Office of Biological and Environmental Research (BER), Environmental
System Science (ESS) Program. A portion of this research was performed at the Environmental Molecular Sciences
Laboratory, a DOE Office of Science User Facility sponsored by the Biological and Environmental Research
program under Contract No. DE-AC05-76RL01830 and user proposal 51180. We thank Sophia McKever for
generating Figure 1.

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

**Figures**

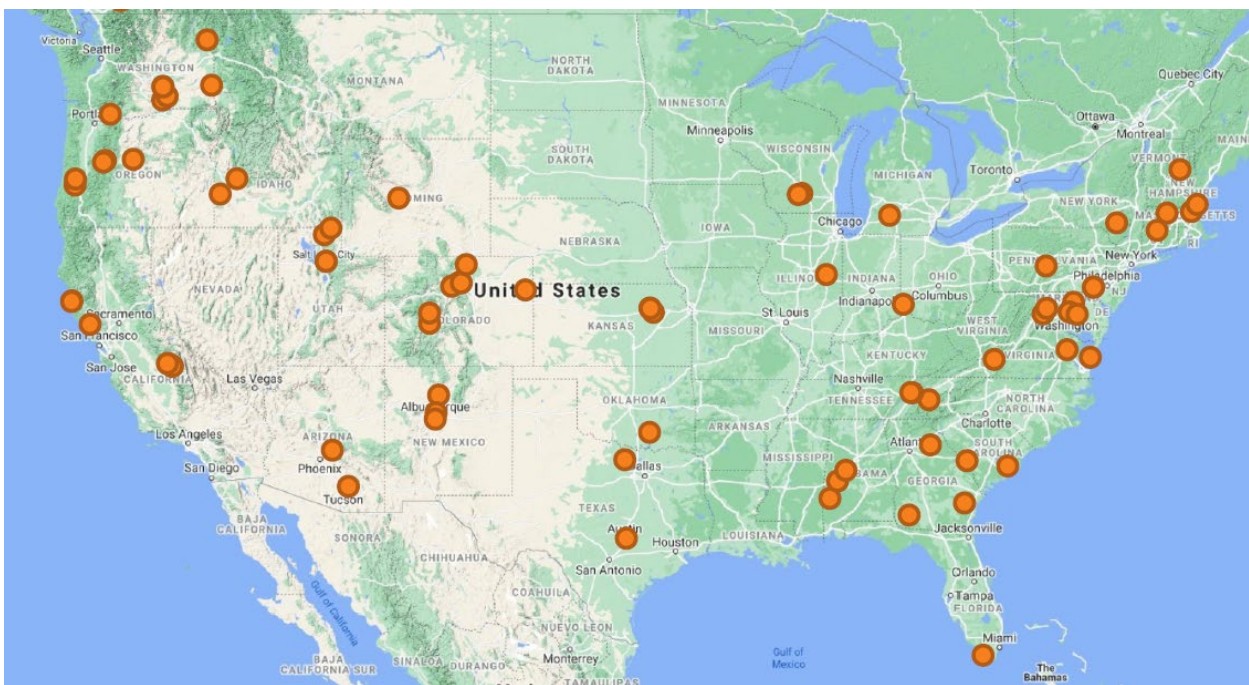


**Figure 1.** Map of sampling locations distributed across the contiguous United States (ConUS). Surface water and
sediments were collected at each site using a crowdsourced approach via the WHONDRS consortium. Physical
factors such as stream order were not constrained. Figure generated by Sophia McKever using QGIS. The base map
is copyrighted: © OpenStreetMap contributors 2022. Distributed under the Open Data Commons Open Database
504 License (ODbL) v1.0.

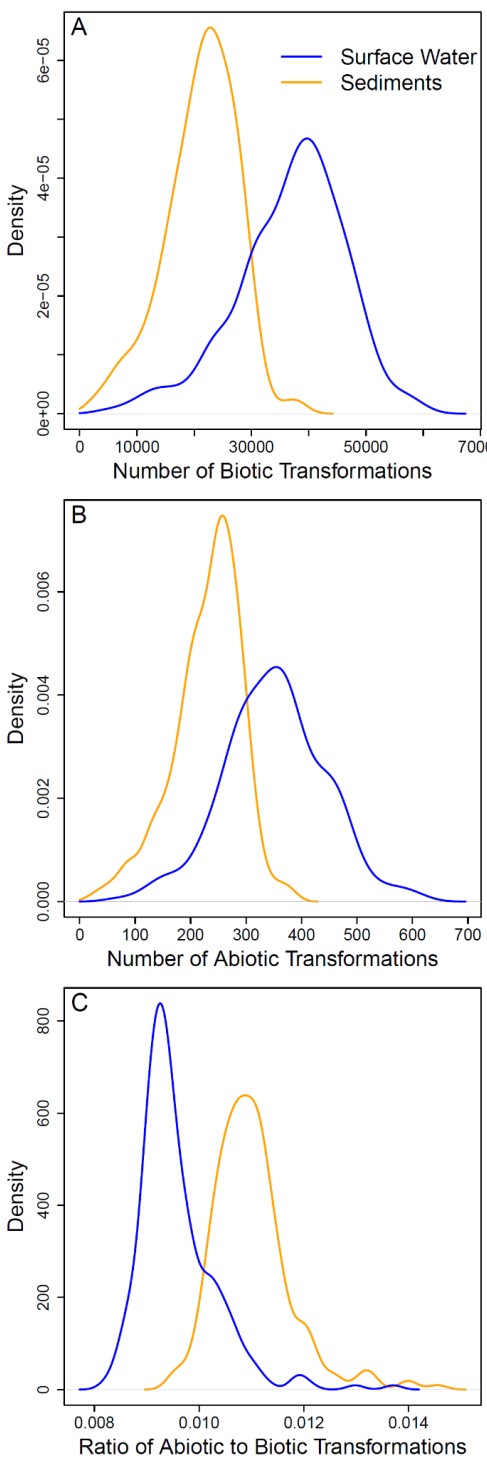


**Figure 2.** Examining the ConUS-scale distributions of biotic and abiotic transformation numbers reveals more
transformations in surface water than sediment organic matter. Kernel density functions for ConUS-scale biotic (A)
and abiotic (B) transformations, and their ratio (C) in sediment (orange lines) and surface water (blue lines) organic
matter. The median values of the distributions significantly diverge within each panel (see text for statistics).

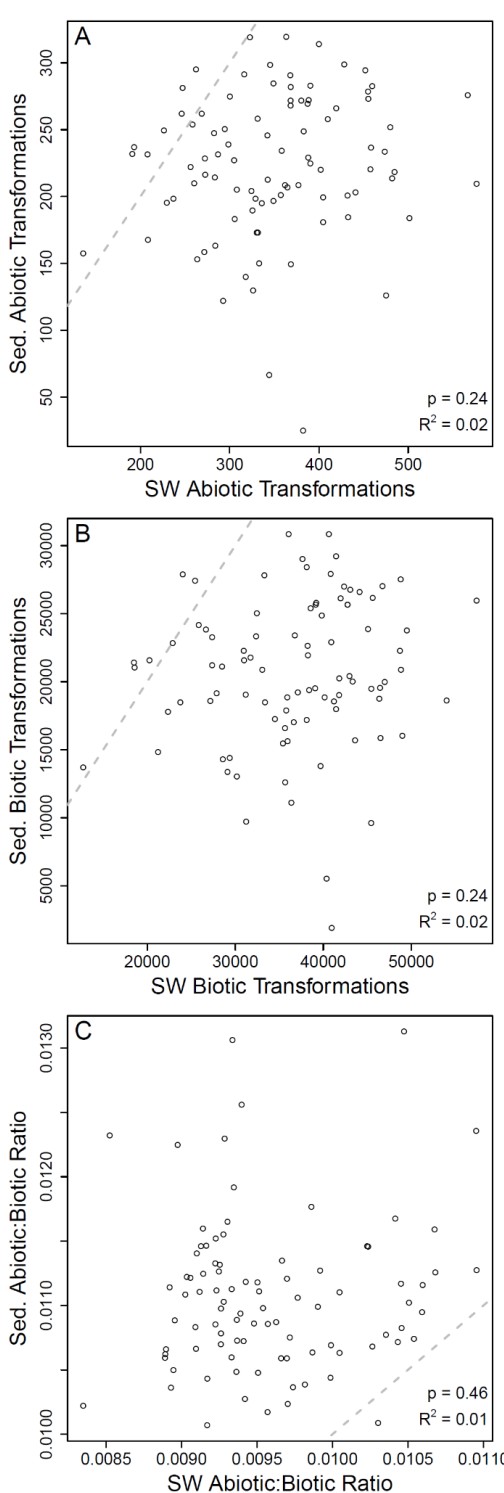

**Figure 3.** Sediment (Sed.) and surface water (SW) transformation counts and are not related to each other.
Regression analysis of the number of abiotic (A) and biotic (B) transformations and their ratio (C). Each open circle
is from one sampling site in which surface water and sediments were both collected. Regression statistics are
provided on each panel and the dashed line is the 1-to-1 line; no regressions were significant.

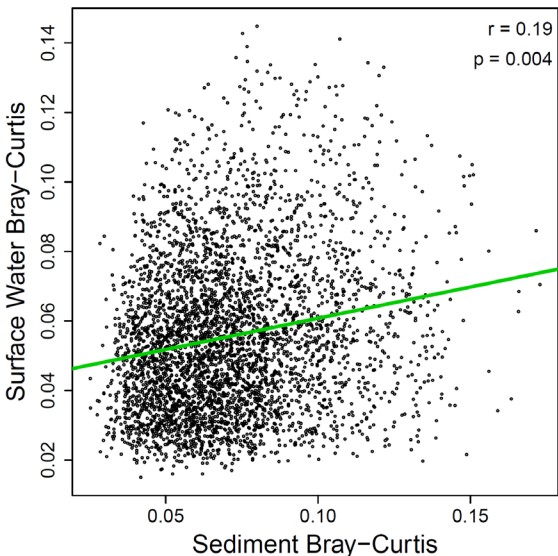


**Figure 4.** Transformation profiles of OM in sediments and surface water were weakly related to each other. Bray-
Curtis dissimilarities in surface water and sediments are plotted against each other, with their relationship evaluated
via Mantel test to control for non-independence among data points (see Methods). The Pearson correlation
coefficient and the Mantel-based p-value are provided on the panel. While significant, the relationship is extremely
weak, suggesting lack of a meaningful relationship. One outlier sample was discovered and excluded from this
analysis. Figure S1 includes the outlier, which does not change the interpretation, it only makes it harder to see the
data.

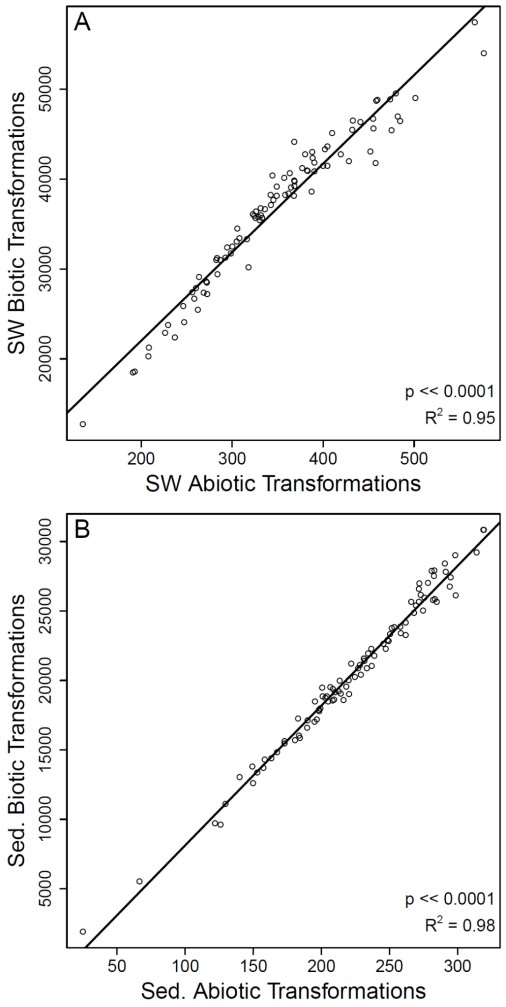


**Figure 5.** Strong correlations were observed between the number of biotic and abiotic organic matter transformations within surface water (SW) and within sediment (Sed.). Each circle represents one sampled site for surface water (A) and sediments (B). The solid black line is the regression model and statistics are provided on each panel.
