# Peer review of "Organic Matter Transformations are Disconnected Between Surface Water and the Hyporheic Zone"

_Biogeosciences, 2022_

## Author Comment (AC1)

**Reviewer 1:**

General comments

From an FTICR-MS formula data set of sediment and surface water samples biogeochemical transformations were derived and compared. Mass shifts in the mass spectra of the samples and a known list of metabolic relevant mass differences were used. The main outcome was that surface water DOM transformations are driven by upstream catchment processes and hyporheic zone transformations local to the sample volume.

DOM cycling is of biogeochemical importance. The data evaluation and the hypotheses made are plausible. The study is so far well structured and understandable. Recent and appropriate literature was used for discussion. This indicates that the present study is appropriate to Biogeosciences.

Thank you for the encouraging evaluation. Please see our thoughts related to your comments below.

Detail comments

Lines 32 – 34: Why is it surprising that DOM transformation processes are different from those in sediment or sediment pore-water? One can suggest that redox conditions and many other parameters are highly different as such nutrient (N, P) availability. In addition one can expect that processes must be different. In sediments and hyporheic zone adsorption / desorption processes (with participation of iron and other minerals) may play a role. In surface water photochemical transformations are possible whereas in sediment are of minor importance. In addition in surface water and sediment pore water different DOM species may be dominant driving other processes.

The text will be edited to more concisely indicate that mixing does not overcome differences between water and sediments in organic matter production, transformation, and/or sorption/desorption processes.

Lines 68 – 74: Here it is convincingly argued that photoproducts (which can be available) produced at the water surface can via hydrologic mixing enhance microbial turnover in the hyporheic zone.

Thank you.

Line 114: How many mL of filtrated supernatant sediment water extract were passed through which PPL cartridges (50 mg, 100 mg, 500 mg…)?

The method description will be expanded to include a volume description of the normalized acidified supernatant passed through the PPL cartridges.

Line 117 – 123: Please provide at least the elements allowed to the mass calculator (how many N, S, P) and give basic information about formula assignment or cite software use, for example: Fu, Q. L.;  Fujii, M.; Riedel, T., Development and comparison of formula assignment algorithms for ultrahigh-resolution mass spectra of natural organic matter. Anal. Chim. Acta 2020, 1125, 247-257.

We agree that the description of spectra processing in the methods section could have additional details regarding formula assignment and references.

Line 137 – 138: list of common biochemical transformations, the associated data package, please refer to Table S3 there. The readership should have the opportunity to immediately find it.

The reference to S3 will be added into the main text.

Please provide a list of all abbreviations used in Tables S1 – S5 (may be in an additional table there)

 We will generate a list to describe the abbreviations used in the S1-S5 tables.

Line 140: this is an interesting idea to regard mass shifts between components as transformation. From my viewpoint one can support the hypothesis that such transformation must have occurred. However there is no reference system to be sure at what time or at what place this transformation did occur. It might have occurred randomly in the past anywhere in the soil or catchment area.

This is a valid point. We consider the mass shifts to indicate potential or putative transformations. We will add text indicating this important caveat to this subsection ("Biochemical transformation analyses and statistics").

In any case the limitations of the mass shifts assumptions as transformation should be discussed.

We agree and in addition to providing a caveat in the Methods subsection (as noted just above), we will revisit this point/caveat in the Results and Discussion. This will be done to (1) emphasize that we are studying putative transformations and (2) such transformations could have occurred in another location and at a previous time, relative to the location and timing of sample collection. This aligns well with our inference that the transformations inferred in surface water reflect processes occurring throughout the upstream catchment.

Line 150: the composition of transformations, is biotic and abiotic meant, and the list in Table S3? Is it possible to mark in an additional column which fragment is suggested biotic and which abiotic?

Our approach for providing all data, metadata, and scripts is via the published data package that is linked in the 'Data Availability' section. That data package has complete data, metadata, scripts, and file-level metadata. We would greatly prefer to stick with this strategy as the data package is well-structured, documented, publicly available, and citable. The specific file asked for in this comment is in that data package as well. It is the file named: Biotic-abiotic-transformations-classification.csv.

Question: Only the mass shifts were evaluated, not the mass peak intensities? Is this the reason why the intensities were not provided in Table S2?

For this analysis we did not use peak intensities such that we feel it is best to not provide that information. The full dataset with peak intensities is available from the published data package we used, however. We chose to not use peak intensities because in FTICR-MS data, differences in peak intensities do not reflect differences in concentration. In turn, it is unclear how to interpret differences in peak intensities. This is because peak intensity of any given molecule in a given sample is based on the ionization efficiency of that molecule which in turn is heavily influenced by which other molecules are present and detailed properties of the molecule's physics and how it responds to ionization.

Line 184 – 185: As a limitation of this statement, only the number of mass shifts was evaluated. There is no information if Y shifts came from leaching the sediment and X additional shifts came from further reactions in the pore-water. Biogeochemically active means this location where the sample was taken from, not the place from where the sample composition was generated and afterwards transported to the location under consideration.

We would be happy to consider adding an additional caveat to this statement, though we are a little unclear how to modify the text. The sentence is setup as a simple initial hypothesis. That is, because hyporheic zone sediments are often more biogeochemically active than surface water, we might expect that

Line 188 – 192: to mention this limitation, "not the rate of transformation", is very important here. I applaud all the limitations mentioned here.

Thank you.

Line 197: the accumulation of transformations (larger diversity) is a convincing hypothesis.

Thank you.

Line 269 – 271: as abiotic transformations, adsorption / desorption should be taken into account besides redox reactions

We agree and will add some additional text to address adsorption/desorption chemistry in the context of abiotic transformation in sediments.

**Reviewer 2:**

In the submitted work, the authors compared the number of biotic and abiotic transformations occurring between river DOM and hyporheic zone sediment DOM to investigate the OM transformation and chemical differences/similarities in the two DOM groups in the hydrologic connctivity. The number of the transforamtions was counted by molecular-level composition of DOM analyzed by FT-ICR-MS. The study is very interesting. However, there are several concerns/questions about the methodology and the arguments in the manuscript.

Thank you for the positive feedback, and please see our thoughts on your comments below.

(1) As the authors mentioned, FT-ICR-MS has its inherent limitations due to the short range of MW (200-1000 Da) and ionization effects. Therefore, the number of the transformations inferrred from the measured chemical composition may not represent all that might occur in the total DOM pool of samples.  The authors need to state this limitation and potential changes in the conclusions/arguments.

We agree and will add some text to the Conclusions section that directly addresses this important caveat.

(2) If the previous concern could be well resolved, it would be OK to compare the number of transformations between river and sediment DOM. However, I am still not sure about that between abiotic and biotic transformation in a given DOM group because some biotic (or abiotic) transformation processes may depend more on the limited analytical window of FT-ICR-MS than others.

This is highly related to the comment just above, and we plan to address this caveat in the Conclusion section as well. More specifically, we will state that all results and inferences in our paper are inherently limited to the analytical window of the FTICR-MS instrument. The FTICR-MS is the most powerful instrument available in terms of being able to resolve far more organic molecules than other technologies (e.g., LC-MS, GC-MS, NMR). However, every technique has analytical window limitations. We will suggest that follow-on studies could combine data from multiple platforms to provide a more comprehensive characterization of DOM. However, because of its higher resolving power, the FTICR-MS data will likely dominate any merged datasets such that would hypothesize that the patterns observed in our study would likely be maintained.

(3) River DOM compositon is affected by diverse sources with different chemical composition, while sediment DOM can relatively homogeneous because of infiltration effect in sub-surface followed by strong interactions with mineral surfaces. Please compare the chemical diversity of DOM between the two groups and add this aspect in the discussion, if acceptable.

DOM chemical diversity can be measured in numerous ways and given that the focus of the paper isn't on chemical diversity per se, we would prefer to keep an analysis of chemical diversity as simple as possible. The simplest evaluation of chemical diversity in FTICR-MS data is the number of unique molecules observed in each sample. We refer to this as 'molecular richness.' Surface water DOM and sediment-associated DOM each have a distribution of molecular richness with a mean, median, etc. Those distributions can be statistically compared to ask whether there is a significant difference between surface water and sediments, in terms of their DOM molecular richness. Using the same dataset, Garayburu-Caruso et al. (2021) found higher molecular richness in surface water DOM, relative to sediment DOM (see their Fig. 1, bottom panel labeled as 'metabolite count' which is the same as 'molecular richness'). We referenced this result on line 199 of the manuscript and used it to help develop a post hoc hypothesis:

"That is, a larger diversity of transformations may accumulate as surface water OM integrates processes and sources from across the stream network, which is conceptually consistent with previous work using the same data that found higher molecular richness in surface water than in sediment OM (Garayburu-Caruso et al., 2020b). This interpretation sets up the emergent (i.e., post-hoc) hypothesis that the number of transformations may increase with catchment area."

Given that we used previous DOM chemical diversity analyses to help develop a new hypothesis in the Results and Discussion section of the manuscript, we would prefer to leave it as-is. We feel that adding additional analyses of DOM chemical diversity will cause the paper to head in a different conceptual direction, relative to its intended goals. Because DOM chemical diversity can be studied in myriad ways, we prefer to hold those deeper analyses for a future manuscript focused on diversity.

As a minor commnet, I suggest the text in line 97-99 to be removed or moved someehter else because the conclusive remark is not appropriate in the introduction section.

We feel this is a stylistic choice in which we are giving the reader the key outcome of the paper at the end of the Introduction before they head into the rest of the paper. The idea is that the reader can have the outcome in their minds as they read through the paper so they can decide for themselves whether that outcome is supported by the results, interpretations, and methods. We prefer to keep this text, if it is acceptable to the reviewer and editor. For easy reference, here is the text: "One key outcome of our analyses is that OM transformations in sediments are not related to OM transformations in adjacent surface water, which suggests divergent governing processes despite hydrologic connectivity between these river corridor sub-systems."

---

## Author Response (AR1)

Dear Dr. Park,

Thank you for securing two high quality reviews and for your own evaluation of our work. We appreciate everyone's suggestions and have followed them to generate an improved manuscript. Below you will find details of our edits and response to each suggestion from yourself and the two reviewers. Our responses are indicated as blue text.

We look forward to your further evaluation.

Thank you,

James Stegen (on behalf of all co-authors)

**#####**

**Editor's Letter:**

Dear Dr. Stegen:

Thank you for providing detailed responses to the comments and suggestions offered by two referees.

The reviewers recognized the novelty and significance of your research. Based on the overall positive evaluations of the reviewers and your thoughtful responses to the relatively small number of correction requirements, I am pleased to recommend 'Publish subject to minor revisions'.

Thank you, this is exciting news, and we greatly appreciate your efforts and those of the reviewers in helping to improve the manuscript.

Regarding the third comment from the second reviewer ("River DOM composition is affected by diverse sources with different chemical composition, while sediment DOM can be relatively homogeneous because of infiltration effect in sub-surface followed by strong interactions with mineral surfaces. Please compare the chemical diversity of DOM between the two groups and add this aspect in the discussion, if acceptable."), I also think that a short discussion of the homogenizing effect of DOM-mineral interactions would provide readers with a more balanced view of your claim on the "decoupling between surface water and sediment OM". Please conder to provide a short discussion of OM transformations on sediment mineral surface by referring to some relevant previous studies focusing on OM-mineral interactions.

We added a brief discussion on this point and provided associated citations in the 6th paragraph of the Results and Discussion.

I would like to ask you to make all the changes easily identifiable in a marked-up manuscript based on your point-by-point responses to the reviewer comments. If possible, please specify the line numbers of the revised parts in your responses accompanying the revised manuscript.

We have provided a revised manuscript with all changes tracked. We elected to indicate the paragraph numbers as the line numbering can sometimes be off depending on formatting and whether tracked changes are being viewed or not. Please let us know if you prefer a different approach.

Sincerely,

Ji-Hyung Park

Associate Editor, Biogeosciences

**Reviewer 1:**

General comments

From an FTICR-MS formula data set of sediment and surface water samples biogeochemical transformations were derived and compared. Mass shifts in the mass spectra of the samples and a known list of metabolic relevant mass differences were used. The main outcome was that surface water DOM transformations are driven by upstream catchment processes and hyporheic zone transformations local to the sample volume.

DOM cycling is of biogeochemical importance. The data evaluation and the hypotheses made are plausible. The study is so far well structured and understandable. Recent and appropriate literature was used for discussion. This indicates that the present study is appropriate to Biogeosciences.

Thank you for the encouraging evaluation. Please see our thoughts related to your comments below.

Detail comments

Lines 32 – 34: Why is it surprising that DOM transformation processes are different from those in sediment or sediment pore-water? One can suggest that redox conditions and many other parameters are highly different as such nutrient (N, P) availability. In addition one can expect that processes must be different. In sediments and hyporheic zone adsorption / desorption processes (with participation of iron and other minerals) may play a role. In surface water photochemical transformations are possible whereas in sediment are

of minor importance. In addition in surface water and sediment pore water different DOM species may be dominant driving other processes.

The text has been edited to more concisely indicate that mixing does not overcome differences between water and sediments in terms of the processes that influenced organic matter. This is near the end of the Abstract.

Lines 68 – 74: Here it is convincingly argued that photoproducts (which can be available) produced at the water surface can via hydrologic mixing enhance microbial turnover in the hyporheic zone.

 Thank you.

Line 114: How many mL of filtrated supernatant sediment water extract were passed through which PPL cartridges (50 mg, 100 mg, 500 mg…)?

We now include the volume of sample passed through the PPL cartridge. This is in the middle of the first paragraph of the Methods section.

Line 117 – 123: Please provide at least the elements allowed to the mass calculator (how many N, S, P) and give basic information about formula assignment or cite software use, for example: Fu, Q. L.;  Fujii, M.; Riedel, T., Development and comparison of formula assignment algorithms for ultrahigh-resolution mass spectra of natural organic matter. Anal. Chim. Acta 2020, 1125, 247-257.

These details have been added at the end of the first paragraph of the Methods section.

Line 137 – 138: list of common biochemical transformations, the associated data package, please refer to Table S3 there. The readership should have the opportunity to immediately find it.

The list of biochemical transformations are provided in the public data package. We edited the text to indicate the specific file within the data package that has the list of biochemical transformations..

Please provide a list of all abbreviations used in Tables S1 – S5 (may be in an additional table there)

While the manuscript does not contain supplementary tables, we generated a supplementary table that includes all acronyms and abbreviations used across the entire manuscript. It is cited in the supplementary figure file.

Line 140: this is an interesting idea to regard mass shifts between components as transformation. From my viewpoint one can support the hypothesis that such transformation must have occurred. However there is no reference system to be sure at what time or at

what place this transformation did occur. It might have occurred randomly in the past anywhere in the soil or catchment area.

We added text to the first paragraph of the "Biochemical transformation analyses and statistics" sub-section to point out that inferred transformations likely occurred outside of the sample volume.

In any case the limitations of the mass shifts assumptions as transformation should be discussed.

Throughout the Results and Discussion section we included 'putative' in a number of locations to highlight that the inferred transformations are putative. In the third paragraph of the Results and Discussion we added text to again emphasize that inferred transformations likely occurred in another location and at a previous time, relative to the location and timing of sample collection.

Line 150: the composition of transformations, is biotic and abiotic meant, and the list in Table S3? Is it possible to mark in an additional column which fragment is suggested biotic and which abiotic?

Our approach for providing all data, metadata, and scripts is via the published data package that is now cited in the Methods section and is cited in the 'Data Availability' section. That data package has complete data, metadata, scripts, and file-level metadata. We would greatly prefer to stick with this strategy as the data package is well-structured, documented, publicly available, and citable. The specific file asked for in this comment is in that data package as well. It is the file named: Biotic-abiotic-transformations-classification.csv.

Question: Only the mass shifts were evaluated, not the mass peak intensities? Is this the reason why the intensities were not provided in Table S2?

For this analysis we did not use peak intensities such that we feel it is best to not provide that information. The full dataset with peak intensities is available from the published data package we used, however. We chose to not use peak intensities because in FTICR-MS data, differences in peak intensities do not reflect differences in concentration. In turn, it is unclear how to interpret differences in peak intensities. This is because peak intensity of any given molecule in a given sample is based on the ionization efficiency of that molecule which in turn is heavily influenced by which other molecules are present and detailed properties of the molecule's physics and how it responds to ionization.

Line 184 – 185: As a limitation of this statement, only the number of mass shifts was evaluated. There is no information if Y shifts came from leaching the sediment and X additional shifts came from further reactions in the pore-water. Biogeochemically active means this location where the sample was taken from, not the place from where the sample composition was generated and afterwards transported to the location under consideration.

We would be happy to consider adding an additional caveat to this statement, though we are a little unclear how to modify the text. The sentence is setup as a simple initial hypothesis. That is, because hyporheic zone sediments are often more biogeochemically active than surface water, we might expect there to be more transformations in sediments.

Line 188 – 192: to mention this limitation, "not the rate of transformation", is very important here. I applaud all the limitations mentioned here.

Thank you.

Line 197: the accumulation of transformations (larger diversity) is a convincing hypothesis.

Thank you.

Line 269 – 271: as abiotic transformations, adsorption / desorption should be taken into account besides redox reactions

We added some text to the 8th paragraph of the Results and Discussion related to this point. We kept this next text short as the idea is speculative.

**Reviewer 2:**

In the submitted work, the authors compared the number of biotic and abiotic transformations occurring between river DOM and hyporheic zone sediment DOM to investigate the OM transformation and chemical differences/similarities in the two DOM groups in the hydrologic connctivity. The number of the transforamtions was counted by molecular-level composition of DOM analyzed by FT-ICR-MS. The study is very interesting. However, there are several concerns/questions about the methodology and the arguments in the manuscript.

Thank you for the positive feedback, and please see our thoughts on your comments below.

(1) As the authors mentioned, FT-ICR-MS has its inherent limitations due to the short range of MW (200-1000 Da) and ionization effects. Therefore, the number of the transformations inferrred from the measured chemical composition may not represent all that might occur in the total DOM pool of samples. The authors need to state this limitation and potential changes in the conclusions/arguments.

We added text discussing these caveats to the Conclusions section.

(2) If the previous concern could be well resolved, it would be OK to compare the number of transformations between river and sediment DOM. However, I am still not sure about that between abiotic and biotic transformation in a given DOM group because some biotic (or abiotic) transformation processes may depend more on the limited analytical window of FT-ICR-MS than others.

This is highly related to the comment just above, and the text we added to the Conclusion is meant to address this. More specifically, we state that our inferences are limited to the analytical limits of the FTICR-MS data. The FTICR-MS is the most powerful instrument available in terms of being able to resolve far more organic molecules than other technologies (e.g., LC-MS, GC-MS, NMR). However, every technique has analytical window limitations. The revised text calls this out and it encourages follow-on studies that combine data from multiple platforms to provide a more comprehensive characterization of DOM. However, because of its higher resolving power, the FTICR-MS data will likely dominate any merged datasets such that would hypothesize that the patterns observed in our study would likely be maintained; this is also noted in the revised text

(3) River DOM compositon is affected by diverse sources with different chemical composition, while sediment DOM can relatively homogeneous because of infiltration effect in sub-surface followed by strong interactions with mineral surfaces. Please compare the chemical diversity of DOM between the two groups and add this aspect in the discussion, if acceptable.

DOM chemical diversity can be measured in numerous ways and given that the focus of the paper isn't on chemical diversity per se, we would prefer to keep an analysis of chemical diversity as simple as possible. The simplest evaluation of chemical diversity in FTICR-MS data is the number of unique molecules observed in each sample. We refer to this as 'molecular richness.' Surface water DOM and sediment-associated DOM each have a distribution of molecular richness with a mean, median, etc. Those distributions can be statistically compared to ask whether there is a significant difference between surface water and sediments, in terms of their DOM molecular richness. Using the same dataset, Garayburu-Caruso et al. (2021) found higher molecular richness in surface water DOM, relative to sediment DOM (see their Fig. 1, bottom panel labeled as 'metabolite count' which is the same as 'molecular richness'). We referenced this result on line 199 of the original manuscript and used it to help develop a post hoc hypothesis. That text is provide here:

"That is, a larger diversity of transformations may accumulate as surface water OM integrates processes and sources from across the stream network, which is conceptually consistent with previous work using the same data that found higher molecular richness in surface water than in sediment OM (Garayburu-Caruso et al., 2020b). This interpretation sets up the emergent (i.e., post-hoc) hypothesis that the number of transformations may increase with catchment area."

Given that we used previous DOM chemical diversity analyses to help develop a new hypothesis in the Results and Discussion section of the manuscript, we would prefer to leave it as-is. We feel that adding additional analyses of DOM chemical diversity will cause the paper to head in a different conceptual direction, relative to its intended goals. Because DOM chemical diversity can be studied in myriad ways, we prefer to hold those deeper analyses for a future manuscript focused on diversity.

As a minor commnet, I suggest the text in line 97-99 to be removed or moved someehter else because the conclusive remark is not appropriate in the introduction section.

We feel this is a stylistic choice in which we are giving the reader the key outcome of the paper at the end of the Introduction before they head into the rest of the paper. The idea is that the reader can have the outcome in their minds as they read through the paper so they can decide for themselves whether that outcome is supported by the results, interpretations, and methods. We prefer to keep this text, if it is acceptable to the reviewer and editor. For easy reference, here is the text: "One key outcome of our analyses is that OM transformations in sediments are not related to OM transformations in adjacent surface water, which suggests divergent governing processes despite hydrologic connectivity between these river corridor sub-systems."

Thank you again to Dr. Park and both reviewers for all the helpful suggestions.

---

## Author Response (AR2)

Dear Editor,

No changes were requested, and none have been made.

Sincerely,

James Stegen (on behalf of all co-authors)